# Hybrid optimization between iterative and network fine-tuning reconstructions for fast quantitative susceptibility mapping

**Jinwei Zhang**[1,2]                                    JZ853@CORNELL.EDU
[1] *Department of Biomedical Engineering, Cornell University, Ithaca, NY, USA*
[2] *Department of Radiology, Weill Medical College of Cornell University, New York, NY, USA*

**Hang Zhang**[2,3]                                     HZ459@CORNELL.EDU
[3] *Department of Electrical and Computer Engineering, Cornell University, Ithaca, NY, USA*

**Pascal Spincemaille**[2]                           PAS2018@MED.CORNELL.EDU
**Thanh Nguyen**[2]                                  TDN2001@MED.CORNELL.EDU
**Mert Sabuncu**[1,2,3]                                MSABUNCU@CORNELL.EDU
**Yi Wang**[1,2]                                     YIWANG@MED.CORNELL.EDU

**Editors:** Under Review for MIDL 2021

## Abstract

A Hybrid Optimization Between Iterative and network fine-Tuning (HOBIT) reconstruction method is proposed to solve quantitative susceptibility mapping (QSM) inverse problem in MRI. In HOBIT, a convolutional neural network (CNN) is first trained on healthy subjects' data with gold standard labels. Domain adaptation to patients' data with hemorrhagic lesions is then deployed by minimizing fidelity loss on the patient training dataset. During test time, a fidelity loss is imposed on each patient test case, where alternating direction method of multiplier (ADMM) is used to split the time consuming fidelity imposed network update into iterative reconstruction and network update subproblems alternatively in ADMM, and only a subnet of the pre-trained CNN is updated during the process. Compared to the method FINE where such fidelity imposing strategy was initially proposed to solve QSM, HOBIT achieved both performance gain of reconstruction accuracy and vast reduction of computational time. Our code is available at https://github.com/Jinwei1209/HOBIT.

**Keywords:** convolutional neural network, alternating direction method of multiplier, domain adaptation, quantitative susceptibility mapping

## 1. Introduction

Quantitative susceptibility mapping (QSM) is an imaging contrast in magnetic resonance imaging (MRI) to quantify tissue magnetic susceptibility values from estimated local tissue field data (Kressler et al., 2009). QSM provides biomarkers for tissues with iron, calcium and gadolinium (Wang and Liu, 2015) concentrations which can be used for clinical diagnosis, such as multiple sclerosis (Langkammer et al., 2013), intracranial calcifications and hemorrhages (Chen et al., 2014). QSM is computed by inverting the following forward process:

$$b = F^H DF\chi + n \tag{1}$$

where $b$ is the estimated local tissue field from magnetic resonance phase imaging, $\chi$ is the tissue susceptibility to compute, $F$ is the Fourier transform, $D$ is the dipole kernel in k-space and $n$ is the additive noise (assuming i.i.d. Gaussian for each voxel). With single orientation sampling, the dipole inversion problem from local field $b$ to susceptibility $\chi$ is ill-posed since the zero-cone in the k-space dipole kernel produces dipole-incompatible field, which results in streaking and shadow artifacts of susceptibility (Kee et al., 2017).

Various methods have been proposed to resolve the ill-posedness of dipole inversion. Direct method truncated k-space division (TKD) modified the dipole kernel near the zero-cone to add dipole-incompatible field components (Shmueli et al., 2009). Iterative method morphology enabled dipole inversion (MEDI) introduced a weighted total variation regularization to suppress the streaking artifacts (Liu et al., 2012). Oversampling method calculation of susceptibility through multiple orientation sampling (COSMOS) eliminated the zero-cone of the dipole kernel by a combination of multi-oriented fields (Liu et al., 2009); therefore, COSMOS has been regarded as the gold standard susceptibility map. With the advance of convolutional neural network (CNN), deep learning (DL) has been introduced in QSM. A first deep learning method QSMnet built a 3D U-Net for field-to-susceptibility mapping using COSMOS as the training dataset, and was reported to porform better than TKD and MEDI (Yoon et al., 2018). Another deep learning method DeepQSM trained U-Net with synthetic field-susceptibility pairs (Bollmann et al., 2019). Since then, more architectures have been proposed based on the backbone U-Net, such as QSMGAN (Chen et al., 2020), xQSM (Gao et al., 2020) and folded attention U-Net (Zhang et al., 2020a) to name a few.

Compared to conventional methods, DL QSM methods usually achieve fast and accurate reconstructions on test dataset, but when tested on the cases with pathologies not encountered during training, such as intracranial calcifications and hemorrhages with extreme susceptibility values, generalization error may be enlarged in those regions. The generalization error could show up as severely under-estimated susceptibility values of lesions in DL QSM. To overcome such limitation, several methods were proposed to improve the domain adaptation ability of DL QSM. QSMnet+ augmented the training dataset to a wider range of susceptibility in order to generalize the network better (Jung et al., 2020). Probabilistic dipole inversion (PDI) adapted the pre-trained network to different patient datasets using variational inference (Zhang et al., 2020d,c). Fidelity imposed network edit (FINE) deployed the fidelity loss of dipole inversion on each test case so that the generalization error of unseen lesions could be reduced (Zhang et al., 2020b).

As one of effective domain adaptation methods for DL QSM, FINE combines advantageous robustness of iterative methods and implicit regularization of DL methods. Despite such merit, significantly increased computational time is needed for FINE, which hinders its practical usage. In the work, we analyze existing issues of FINE and attempt to resolve them all with a newly proposed method derived from FINE: Hybrid Optimization Between Iterative and network fine-Tuning (HOBIT) reconstruction for fast QSM. We deployed ablation study of HOBIT and compared it with MEDI, QSMnet, QSMnet+, FINE and PDI on both in vivo and simulated hemorrhagic datasets. Superior reconstruction performance was achieved in HOBIT and reconstruction speed was vastly accelerated compared to FINE.

## 2. Method

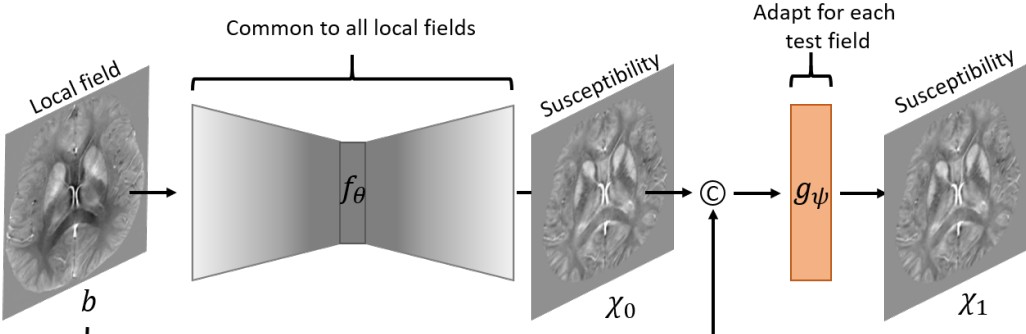

Figure 1: Network architecture in HOBIT. $f_\theta$ was the dipole inversion network 3D U-Net and $g_\psi$ was a slimmer network with five convolutional layers. $f_\theta$ has a single input $b$ while $g_\psi$ has $b$ and $f'_\theta s$ output $\chi_0$ concatenating together as its input to produce the final output $\chi_1$. Only $g_\psi$ is adapted for each test case after training.

### 2.1. Issues in FINE

In FINE (Zhang et al., 2020b), a 3D U-Net (Ronneberger et al., 2015) was pre-trained on the multi-orientation dataset of healthy subjects with COSMOS (Liu et al., 2009) as labels to do supervised learning. When tested on each patient data without label, FINE adapted the pre-trained weights by minimizing the following fidelity loss in an unsupervised fashion:

$$\|W(F^H D F \chi - b)\|_2^2 \tag{2}$$

until the relative change of fidelity loss between two consecutive epochs fell below $5 \times e^{-3}$, where $W$ is the square root of the inverse of the noise covariance matrix. The vanilla FINE above has three major issues:

- When performing FINE in subject, pathology-related domain adaptation information is not inherited when performing FINE in other subjects that have a similar pathology.

- The whole network update of FINE is redundant, as lots of weights seldom change during network update (Fig. 2 in FINE (Zhang et al., 2020b)).

- Network update leads to slow update of the output susceptibility, requiring hundreds of epochs to converge.

In the next section, we attempt to tackle the three issues above using the proposed method.

### 2.2. HOBIT

In HOBIT, we design the network architecture as shown in Fig. 1, where a first dipole inversion network 3D U-Net $f_\theta$ maps local field input $b$ to susceptibility output $\chi_0$, then a

slimmer network $g_\psi$ consisting of five convolutional layers maps $\{\chi_0, b\}$ to the final suscep-
tibility output $\chi_1$. COSMOS dataset of healthy subjects are used to pre-train $f_\theta$ and $g_\psi$
with the following loss function:

$$\min_{\theta,\psi} \sum_{i=1}^{N_C} \|\chi_0^{(i)} - \chi_T^{(i)}\|_1 + \|\chi_1^{(i)} - \chi_T^{(i)}\|_1, \tag{3}$$

where $\chi_T^{(i)}$ is the $i$-th label/target from a total of $N_C$ COSMOS data points, $\chi_0^{(i)}$ and $\chi_1^{(i)}$
are predictions of $f_\theta$ and $g_\psi$. After pre-training, the following steps are deployed to resolve
the three major issues of FINE described in section 2.1 point-by-point:

- Domain adaptation to the patient dataset is accomplished by fine-tuning the COSMOS
  pre-trained network with a fidelity loss function on the patient training dataset:

$$\min_{\theta,\psi} \sum_{i=1}^{N_P} \|W^{(i)}(F^H DF\chi_0^{(i)} - b^{(i)})\|_2^2 + \|W^{(i)}(F^H DF\chi_1^{(i)} - b^{(i)})\|_2^2, \tag{4}$$

  where $W^{(i)}$ and $b^{(i)}$ are the $i$-th noise weighting matrix and input local field from a
  total of $N_P$ patient data points. Then during test per case, network refinement starts
  from those domain adapted weights.

- After domain adaptation using Eq. 4, $f_\theta$ is fixed and only $g_\psi$ is refined for each test
  case in the patient test dataset.

- Rewrite minimization of network reparametrized fidelity loss $\frac{1}{2}\|W(F^H DF g_\psi(\chi_0, b) - b)\|_2^2$ as:

$$\min_{\psi,\chi} \frac{\alpha}{2}\|W(F^H DF\chi - b)\|_2^2 + \frac{1-\alpha}{2}\|W(F^H DF g_\psi(\chi_0, b) - b)\|_2^2$$
$$\text{s.t. } \chi = g_\psi(\chi_0, b), \tag{5}$$

  where $0 \le \alpha \le 1$. Convert the constrained optimization problem in Eq. 5 as the
  augmented Lagrangian format:

$$\min_{\psi,\chi} \frac{\alpha}{2}\|W(F^H DF\chi - b)\|_2^2 + \frac{1-\alpha}{2}\|W(F^H DF g_\psi(\chi_0, b) - b)\|_2^2$$
$$+ \frac{\rho}{2}\|\chi - g_\psi(\chi_0, b) + \mu\|_2^2 - \frac{\rho}{2}\|\mu\|_2^2, \tag{6}$$

  where $\rho$ is the penalty parameter and $\mu$ is the dual variable. Eq. 6 is then solved using
  alternating direction method of multiplier (ADMM) (Boyd et al., 2011) iteratively in
  three subproblems:

$$\chi^{n+1} = \arg\min_\chi \frac{\alpha}{2}\|W(F^H DF\chi - b)\|_2^2 + \frac{\rho}{2}\|\chi - g_{\psi^n}(\chi_0, b) + \mu^n\|_2^2, \tag{7}$$

$$\psi^{n+1} = \arg\min_\psi \frac{1-\alpha}{2}\|W(F^H DF g_\psi(\chi_0, b) - b)\|_2^2 + \frac{\rho}{2}\|\chi^{n+1} - g_\psi(\chi_0, b) + \mu^n\|_2^2, \tag{8}$$

$$\mu^{n+1} = \mu^n + \chi^{n+1} - g_{\psi^{(n+1)}}(\chi_0, b), \tag{9}$$

where subproblem Eq. 7 is the network output regularized least square problem which can be approximated with a few conjugate gradient (CG) iterations, subproblem Eq. 8 is the L2 regularized nonlinear least square problem with network reparametrization, which can be solved using first order adaptive gradient descent algorithm such as Adam (Kingma and Ba, 2014).

## 3. Experiments

### 3.1. Data acquisition and preprocessing

Multi-echo 3D gradient echo (MGRE) sequence was performed on 7 healthy subjects using a 3T GE scanner with 5 brain orientations, $256 \times 256 \times 48$ acquisition matrix and $1 \times 1 \times 3$ mm$^3$ voxel size. After data acquisition, raw field data of each scan was estimated via non-linear least square fitting of multi-echo phase data using Levenberg–Marquardt algorithm (Liu et al., 2013). Phase wraps of raw field data were unwrapped using graph-cut based spatial phase unwrapping algorithm (Dong et al., 2014). Background field of raw field data was then removed using projection onto dipole fields (Liu et al., 2011) to obtain local tissue field data $b$ as network's input. COSMOS gold standard as pre-training label was computed by aggregating multi-orientated local fields to do dipole inversion (Liu et al., 2009). MGRE sequence was also performed on 7 intracerebral hemorrhagic (ICH) patients with single orientation and same scanning parameters as COSMOS dataset. Image processing procedures as above were deployed on ICH dataset, except for the COSMOS reconstruction step. Data were acquired following an IRB approved protocol.

For COSMOS pre-training in Eq. 3, data of 5/2 subjects (25/10 brain volumes) were used as training/validation datasets with $\pm 15°$ in-plane rotations for augmentation. Brain volumes were divided into 3D patches with patch size $64 \times 64 \times 32$ and extraction step $21 \times 21 \times 11$, generating 12074/5748 patches for training/validation. For ICH patient domain adaptation in Eq. 4, whole brain volume data from 4/1 subjects were used as training/validation datasets. Data from the remaining 2 patients were used as in vivo ICH test dataset. Simulated local fields were also obtained by applying forward model Eq. 1 to FINE reconstructed QSMs of ICH validation and test datasets, where 5 simulated local fields with different samples of Gaussian noise $n$ were generated on each ICH patient, yielding 5/10 volumes as simulated ICH validation/test datasets. The purpose of these simulated ICH datasets was to provide ground truth (GT) for both ablation study on HOBIT and quantitative comparison among different methods. Peak signal-to-noise ratio (PSNR), root-mean-square error (RMSE), structural similarity index measure (SSIM) (Wang et al., 2004), high-frequency error norm (HFEN) (Ravishankar and Bresler, 2010) and shadow artifact quantification metric of ICH ($R_{ICH}$, defined in Appendix) (Liu et al., 2017) were used as quantitative metrics to evaluate reconstruction accuracy.

### 3.2. Implementation details and ablation study

For network training, $f_\theta$ and $g_\psi$ were first trained with loss Eq. 3 on the COSMOS dataset using Adam optimizer (Kingma and Ba, 2014) (learning rate $10^{-3}$, 60 epochs). $f_\theta$ and $g_\psi$ were then adapted to the ICH patient data with loss Eq. 4 on the in vivo ICH dataset using Adam optimizer (learning rate $10^{-3}$, 200 epochs). In HOBIT, the number of outer

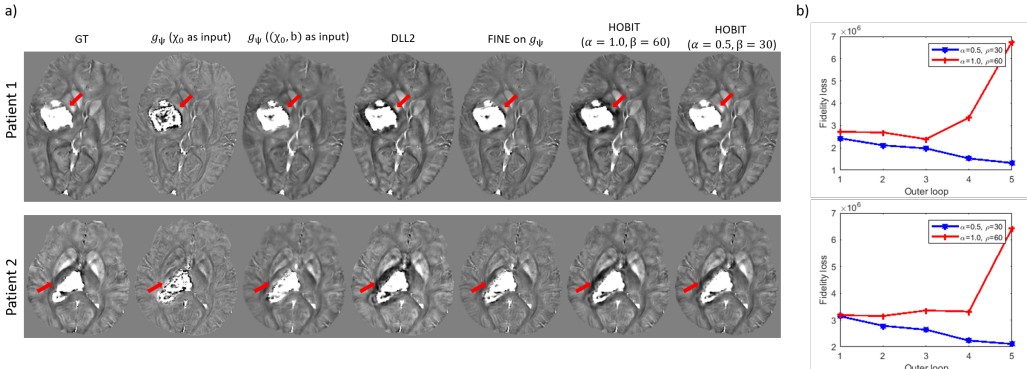

Figure 2: (a): Reconstruction results of two test cases in ablation study ([-0.15, 0.15] ppm). DLL2 and HOBIT ($\alpha = 1.0, \rho = 60$) suffered from shadow artifacts surrounding the hemorrhages (red arrows). (b): Fidelity costs of HOBIT with $\alpha = 0.5$ (monotonically decreasing) and $\alpha = 1.0$ (divergent) per ADMM outer loop.

loops in ADMM was fixed as 5, the relative change threshold of CG in Eq. 7 was $10^{-10}$ with a maximum of 100 iterations, and the number of gradient descent in Eq. 8 was 4 using Adam optimizer (learning rate $10^{-3}$). To determine the optimal $\alpha$ and $\rho$ in Eq. 7 and 8, we applied a grid search of $\alpha$ ([0, 1], interval 0.1) and $\rho$ ([10, 100], interval 10) on the simulated ICH validation dataset, yielding the optimal parameters $\alpha = 0.5$ and $\rho = 30$.

For ablation study, we compared HOBIT against a few methods below on the simulated ICH test dataset. These methods included direct inference of domain adapted $g_\psi$ without and with input local field concatenation (denoted as $g_\psi(\chi_0$ as input) and $g_\psi((\chi_0, b)$ as input)), iterative reconstruction with $g_\psi$ as L2 regularization ($\mu^n = 0$ in Eq. 7, $\alpha = 1, \rho = 60$, denoted as DLL2), FINE on domain adapted $g_\psi$ using fidelity loss Eq. 2 (denoted as FINE on $g_\psi$), and HOBIT with $\alpha = 1.0$ and $\rho = 60$. Reconstruction results of two test cases are shown in Fig. 2a. Quantitative metrics are shown in Table 1. All the methods resolved the under-estimation issue inside the hemorrhagic lesions. Compared to HOBIT with optimal $\alpha = 0.5$ and $\rho = 30$, DLL2 and HOBIT with $\alpha = 1.0$ and $\rho = 60$ suffered from shadow artifacts surrounding the hemorrhagic lesions (red arrows in Fig. 2a), while $g_\psi$ and FINE on $g_\psi$ suffered from sub-optimal reconstruction accuracy. Fidelity costs Eq. 2 of HOBITs with two sets of parameters after each outer loop in ADMM are shown in Fig. 2b. HOBIT with optimal $\alpha = 0.5$ and $\rho = 30$ had monotonically decreasing fidelity cost. In contrast, HOBIT with $\alpha = 1.0$ and $\rho = 60$ suffered from divergence issue of fidelity cost.

### 3.3. Simulated ICH

HOBIT was compared with other dipole inversion methods on the simulated ICH test dataset. Reconstruction results of two test cases are shown in Fig. 3. MEDI reconstructed piecewise constant QSMs which visually looked smooth. QSMnet had under-estimation issue inside the hemorrhages, which was reduced in QSMnet+. Both QSMnet and QSMnet+ had shadow artifact issue surrounding the hemorrhages (red arrow in Fig. 3). FINE, PDI-

Table 1: Average quantitative metrics of 10 test simulated brains reconstructed by different methods in ablation study. Overall, HOBIT ($\alpha = 0.5, \rho = 30$) performed the best.

| | pSNR (dB ↑) | RMSE (% ↓) | SSIM (↑) | HFEN (% ↓) | $R_{ICH}$(%↓) |
|---|---|---|---|---|---|
| $g_\psi$ ($\chi_0$ as input) | 31.63 | 68.28 | 0.9733 | 65.19 | 40.18 |
| $g_\psi$ (($\chi_0, b$) as input) | 33.65 | 57.29 | 0.9765 | 55.51 | 24.80 |
| DLL2 | 37.91 | 35.04 | **0.9854** | **30.84** | 27.66 |
| FINE on $g_\psi$ | 36.64 | 40.80 | 0.9711 | 41.89 | 9.81 |
| HOBIT ($\alpha = 1.0, \rho = 60$) | 35.88 | 44.20 | 0.9834 | 45.78 | 33.57 |
| HOBIT ($\alpha = 0.5, \rho = 30$) | **38.29** | **33.98** | 0.9834 | 32.12 | **7.99** |

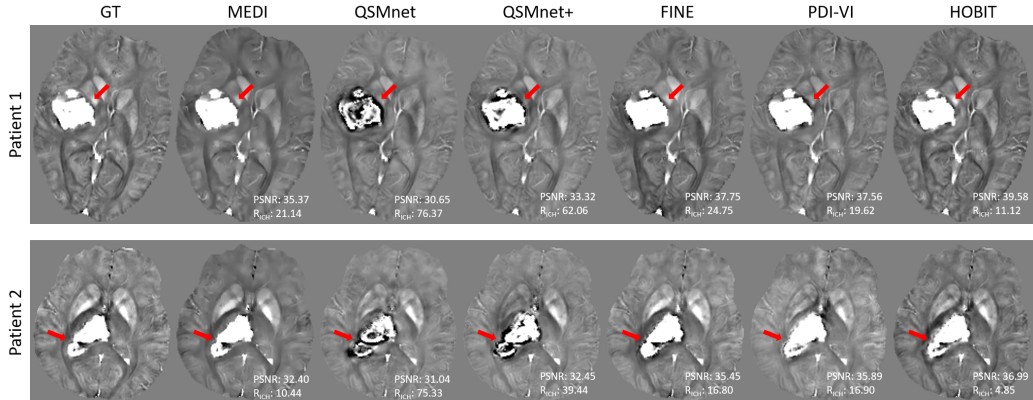

Figure 3: Reconstruction results of two simulated test cases ([-0.15, 0.15] ppm). MEDI visually looked smooth. Under-estimation inside the hemorrhages in QSMnet was reduced in QSMnet+. QSMnet and QSMnet+ had shadow artifact issue surrounding the hemorrhages (red arrows). FINE, PDI-VI and HOBIT produced qualitatively better QSMs than the other methods.

VI and HOBIT produced qualitatively better QSMs than the other methods. Quantitative metrics and computational time of each method are shown in Table 2. HOBIT had the overall best accuracy among all the methods. In terms of computational time per subject, QSMnet, QSMnet+ and PDI achieved the fastest GPU time of less than $1s$, while HOBIT was the fastest iterative method compared to MEDI ($\times 3.1$) and FINE ($\times 31.6$).

## 3.4. In vivo ICH

HOBIT was also compared with other methods on the in vivo ICH test dataset. In this dataset, no ground truth was provided as label; therefore, QSMs were compared qualitatively. Reconstruction results are shown in Fig. 4. Similar to the simulation results in section 3.3, MEDI produced smooth QSMs on the in vivo test data too. QSMnet suffered from under-estimation inside the hemorrhagic lesions while QSMnet+ suffered from severe

Table 2: Average quantitative metrics of 10 simulated ICH test cases. HOBIT achieved the overall best performance and over ×30 faster than FINE on GPU.

|  | pSNR (dB ↑) | RMSE (% ↓) | SSIM (↑) | HFEN (% ↓) | $R_{ICH}$(%↓) | time (s) |
| --- | --- | --- | --- | --- | --- | --- |
| MEDI | 33.89 | 56.62 | **0.9842** | 45.94 | 15.79 | 37.9 |
| QSMnet | 30.85 | 78.96 | 0.9599 | 69.14 | 75.85 | **0.6** |
| QSMnet+ | 32.88 | 62.59 | 0.9794 | 62.47 | 50.75 | **0.6** |
| FINE | 36.60 | 41.17 | 0.9786 | 37.24 | 20.78 | 392.3 |
| PDI-VI | 36.72 | 40.39 | 0.9690 | 41.87 | 18.26 | **0.6** |
| HOBIT | **38.29** | **33.98** | 0.9834 | **32.12** | **7.99** | 12.4 |

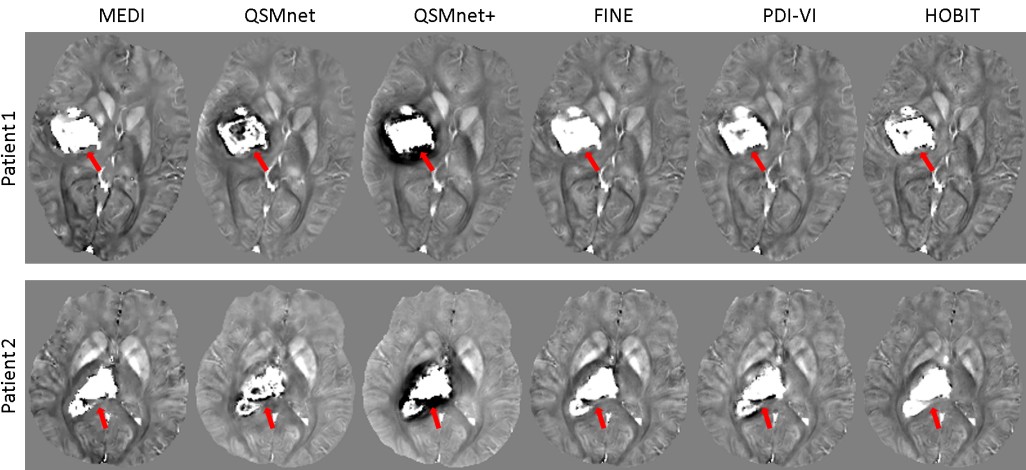

Figure 4: Reconstruction results of two in vivo test cases ([-0.15, 0.15] ppm). MEDI produced smooth QSMs. QSMnet suffered from under-estimation inside the hemorrhagic lesions while QSMnet+ suffered from severe shadow artifacts surrounding the lesions (red arrows). FINE, PDI-VI and HOBIT had visually similar QSMs.

shadow artifacts surrounding the lesions (red arrows in Fig. 4). FINE, PDI-VI and HOBIT had visually similar QSMs including hemorrhages without under-estimation and shadow artifacts and overall susceptibilities without over-smoothness.

## 4. Conclusion

Motivated by analyzing and solving existing issues of FINE, we proposed HOBIT as a novel hybrid iterative and DL reconstruction method for fast QSM. Ablation study showed the necessity of each building block/step in HOBIT for performance improvement. Experiments on both in vivo and simulated ICH test datasets showed that HOBIT achieved over 30 times acceleration on computational time than FINE. Meanwhile, superior reconstruction accuracy was obtained compared to the other methods including FINE.

## Appendix

In this appendix we show the definition of $R_{ICH}$ based on (Liu et al., 2017) to quantify shadow artifact surrounding ICH:

$$R_{ICH} = \big(SD(\chi_{recon}|M_{non-ICH}) - SD(\chi_{GT}|M_{non-ICH})\big)/SD(\chi_{GT}|M_{non-ICH}),$$

where $(\chi|M_{non-ICH})$ denotes the susceptibilities in the non-ICH region defined as 5-mm-wide layer surrounding each ICH, $\chi_{GT}$ denotes ground truth susceptibility, $\chi_{recon}$ denotes reconstructed susceptibility, and $SD(\cdot)$ denotes standard deviation.

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
