# OpenReview forum: "Hybrid optimization between iterative and network fine-tuning reconstructions for fast quantitative susceptibility mapping"
_MIDL.io/2021/Conference — MIDL 2021_

### Official Review · AnonReviewer3 · 2021-03-01

**Confidence:** 4
**Preliminary Rating:** 2
**Final Rating:** 4

**Summary:**

This paper aims to develop a method for domain adaption in QSM through hybrid approach, both utilizing iterative reconstruction and network fine-tuning. Fidelity imposed network edit (FINE), also written by the same group, was one of the pioneering works which aimed at domain adaptation through direct network fine-tuning. The authors give a strong motivation to make the time consuming optimization process faster, and also improve the performance.

**Strengths:**

Overall, the paper is well-written with strong motivation. Problems from the prior work (FINE) are clearly stated, and it is easy to follow how these problems were solved through the newly proposed method, HOBIT. It is fairly easy to follow the method (especially because it was a simple fix), and both retrospective / prospective study were performed to provide backup to the superiority of their work.

**Weaknesses:**

The thing that concerns me the most is the motivation for using both iterative reconstruction and network parameter update. It is clear in the paper **how** these steps are performed, but it is not clear **why** these hybrid optimization steps are beneficial. This concerns me even more since in Table 1, it is observable that HOBIT is not much better than DLL2, which does not update the model parameters but simply does iterative reconstruction. The authors must discuss the merits of the hybrid technique as opposed to iterative reconstruction, and perform experiments that robustifies their claim.

Furthermore, was grid search also performed to find the best parameter when reporting the results for DLL2? If not, grid search should also be performed to give a fair comparison.

**Deanonymize Review:**

no

**Final Rating Justification:**

The authors have fully addressed my concerns by providing the reviewers specific result figures and quantitative metrics. These newly added results are strong evidence that backup their work, and the manuscript is now at an excellent level.

**Justification Of The Preliminary Rating:**

The title of the paper emphasizes that it uses **hybrid technique** for fast QSM. Nonetheless, the manuscript fails to give a comprehensive analysis of the superiority of the hybrid method, hence the rating. If the weaknesses and the questions that I have asked are addressed properly in the rebuttal, I am open to changing the rating.

**Paper Type:**

methodological development

**Questions To Address In The Rebuttal:**

1. What is the motivation for using hybrid technique, instead of iterative reconstruction?

2. Was grid search on parameter selection also performed for DLL2?

**Special Issue:**

no

---

> ### Author Response · Authors · 2021-03-17
> **Motivation for using hybrid technique**
>
> 1: What is the motivation for using hybrid technique, instead of iterative reconstruction?
>
> R3.1: We aim at shortening the computational time of FINE while maintaining its ability to adapt to each test subject through network update. Therefore, we rewrite FINE loss function as Eq. 5 and use ADMM solver to split it into subproblems, where iterative reconstruction subproblem Eq. 7 is fast to compute and network update subproblem Eq. 8 improves network output. In addition, effective domain adaptation pretraining Eq. 4 and slimmer network update $g_{\psi}$ are also used in our study. As a result, we achieve increased computational efficiency and reconstruction accuracy. In contrast, an iterative reconstruction  such as DLL2 potentially reconstructs with inferior image quality if the network output itself is problematic. This issue is reflected in our updated Table 1 (https://drive.google.com/file/d/13sD30Ey-IN90ZTTOZVn7YucgI8f9vrjA/view?usp=sharing) where we add $R_{ICH}$ as a new metric to quantify shadow artifact surrounding ICH. In updated Table 1, DLL2 has larger $R_{ICH}$ (more shadow artifact in Figure 2a: https://drive.google.com/file/d/1YDhVuqFy_eJ-nWVcdJ1Iz8hsQ_HIC66x/view?usp=sharing) than HOBIT (α= 0.5, ρ= 30) since DLL2 uses $g_{\psi}$ as a regularization which already has high $R_{ICH}$. HOBIT attacks this issue using a hybrid technique to correct $g_{\psi}$ in subproblem Eq. 8. Please see our revised paper for a definition of $R_{ICH}$ in the appendix.
>
> 2: Was grid search on parameter selection also performed for DLL2?
>
> R3.2: Yes, DLL2 had similar performance when $\alpha=1.0$ and $40 \leq \rho \leq 80$ (we use $\rho=60$ in the paper), and the R_{ICH} metric was consistently inferior in DLL2 around these parameter range.

---

### Official Review · AnonReviewer1 · 2021-03-05

**Confidence:** 3
**Preliminary Rating:** 4
**Recommendation:** Oral, Poster
**Final Rating:** 4

**Summary:**

The paper proposes a reconstruction method termed HOBIT for quantitative susceptibility mapping (QSM). HOBIT attempts to solve three challenges of the current SOTA model FINE by introducing a pre-training of a U-Net followed by refining a smaller CNN that is adapted based on the test-case to reconstruct. By doing so, domain adaption from healthy volunteer to patient data is achieved. Improved reconstruction performance is achieved compared to several baselines.

**Strengths:**

This paper is well-written. The proposed method nicely introduced, and the evaluation seems solid. The reconstructions are convincing.
-	Good introduction to QSM and literature review
-	Solid description of the method, although far from reproducible (see weakness)
-	Ablation study of the main components of the proposed method
-	Strong evaluation with in vivo patient data as well as simulation data


**Weaknesses:**

The major weakness is that the method is not reproducible. How does the U-Net architecture look like, how does the slimmer network look like? I personally do not need yet another U-Net architecture as a figure in the paper. Simply release the code such that others can apply and compare your method. Also, a proper discussion with limitations and outlook is lacking.

**Deanonymize Review:**

no

**Detailed Comments:**

-	For the statement “The generalization error could show up as severely under-estimated susceptibility values of lesions in DL QSM.”, is there a reference?

Typos:
-	Introduction “porform” -> perform
-	Introduction “in the work” -> this?

**Final Rating Justification:**

The authors addressed my main concern that the method is not reproducible by releasing their code. Thanks.

**Justification Of The Preliminary Rating:**

The paper proposes a method that improves over SOTA DL methods for QSM both in terms of reconstruction performance and speed (cf. FINE). The methodological developments combined with the good evaluation make the paper a very good contribution to MIDL.

**Paper Type:**

both

**Questions To Address In The Rebuttal:**

Please make the code available to foster reproducibility.

**Special Issue:**

no

---

> ### Author Response · Authors · 2021-03-17
> **Source code release**
>
> Our code is available at https://github.com/Jinwei1209/HOBIT. We have included the source code repository link in the abstract section of our revised paper.

---

### Official Review · AnonReviewer2 · 2021-03-08

**Confidence:** 3
**Preliminary Rating:** 3
**Recommendation:** Poster

**Summary:**

The proposed method is an extension of the “fidelity imposed network edit (FINE) for solving the  ill-posed magnetic field to susceptibility map (QSM) inverse problem. The proposed method, HOBIT addresses the existing issues in FINE with respect to domain adaptation, weight update and speed of convergence.

**Strengths:**

The proposed method is not completely supervised and addresses the generalization drawback of supervised learning.

Existing architecture of FINE is modified with a slimmer network and its effect on the overall output is well-projected.

The proposed method inherits the advantages of data-driven approach and domain adaptation to patient data distribution (test time).

The proposed approach is experimented on both simulated and in-vivo ICH test sets.


**Weaknesses:**

In Section 2.1: Issues in FINE, the following sentence says,
“The whole network update of FINE is redundant, as lots of weights seldom change during network update (Fig. 2 in FINE (Zhang et al., 2020b)).”
Specifying Figure 2 here gives an impression to the reader that Figure 2 in the referred paper gives illustrations about weight update. However Figure 2 shows visual comparison and there is no illustration about weight update. Such illustrations are found in Figure 1 of the referred paper. The paper was referred from https://arxiv.org/pdf/1905.07284.pdf

In the baseline method, it is mentioned, “The high-level layers (layers 1 through 3 and layers 16 through 20) of U-Net experienced substantial weight change by FINE.”. Additionally, the base paper mentions, “Our empirical analysis indicates that FINE changes predominately the weights of initial and final (high-level) layers of U-Net for the case of QSM reconstruction in an MS patient (Figure 1), which reflects image contents specific to the patient.”

This means that patient specific contents could potentially be learnt by the higher level layers of the base network also.
However in the proposed method all the layer weights of the FINE network is made fixed, preventing such specific details to be learnt overall. Instead only those layer weights of the FINE which exhibit least Median relative change be fixed and rest be part of the learning process and explore. Accordingly the computational time be analysed.

This sentence is unclear:
After FINE per subject, domain adaptation information is not inherited for FINE on the other cases in the same test dataset.
Patient specific unsupervised learning is done based on the patient-specific data fidelity loss. Hence it is not clear what “other case in the same test dataset” means.

The effect of concatenating the input field with the FINE reconstructed susceptibility is not reflected in the ablative study.

The poor perceptual quality of DLL2 is not in agreement with the highest SSIM score in Table 1. DLL2 exhibits and would mislead the reader in terms of the reconstruction of details.

It is not clear if HFEN is represented as %. If so, it must be mentioned.

FINE, PDI-VI and HOBIT produced qualitatively better QSMs than the other methods. ----- In that case, the PSNR/SSIM metrics for these images should be provided to understand which method performs better.




**Deanonymize Review:**

no

**Detailed Comments:**

In page 2, in the paragraph that starts with “Compared to conventional methods,”, the last two sentences in this paragraph talk about FINE and PDI. To ensure continuity, it would be better to mention about PDI and then lastly talk about FINE, since the following paragraph is about FINE.

In equation 3, the summation should be i=1 to Nc instead of i to Nc. Similar change must be made in equation 4.


**Justification Of The Preliminary Rating:**

The proposed method takes into account the COSMOS dataset for training the baseline network and addresses the additional issues of generalization to real world scenarios especially in cases of scan with abnormalities such as hemorrhage which would cause domain shift between training and test data.

**Paper Type:**

methodological development

**Special Issue:**

no

---

> ### Author Response · Authors · 2021-03-17
> **Response**
>
> 1: Figure 2 confusion in the referred paper.
>
> R1.1: We apologize for the confusion. In this revision, we clarify we are referring to Figure 2 in the published version: https://www.sciencedirect.com/science/article/pii/S1053811920300665, where a weight change was illustrated after FINE in Figure 2.
>
> 2: Subject specific feature learning using $g_{\psi}$.
>
> R1.2: We apologize for the confusion. We now clarify that after training the entire combined network using Eq. 3 and 4, only the first network $f_{\theta}$ in Figure 1 is kept fixed during the hybrid optimization steps in Eq. 7-9, while the second network $g_{\psi}$ is updated. This is similar to the original FINE work, except that the second network was identical in shape to the first network, while in the current work, the second network $g_{\psi}$ is smaller inspired by the observations on what layers were updated in the original FINE work . Since $g_{\psi}$ is designed as five convolutional layers (without downsampling) that are similar to the higher level layers in $f_{\theta}$, we hypothesize that these are sufficient to learn the patient specific changes.
>
> 3: This sentence is unclear: After FINE per subject, domain adaptation information is not inherited for FINE on the other cases in the same test dataset. Patient specific unsupervised learning is done based on the patient-specific data fidelity loss. Hence it is not clear what “other case in the same test dataset” means.
>
> R1.3: We apologize for the confusion. This sentence now reads: “when performing FINE in subject, pathology-related domain adaptation information is not inherited when performing FINE in other subjects that have a similar pathology.”
>
> 4: The effect of concatenating the input field with the FINE reconstructed susceptibility is not reflected in the ablative study.
>
> R1.4: We now add this concatenation effect in the ablation study. Please see our updated Figure 2 (https://drive.google.com/file/d/1YDhVuqFy_eJ-nWVcdJ1Iz8hsQ_HIC66x/view?usp=sharing) and Table 1 (https://drive.google.com/file/d/13sD30Ey-IN90ZTTOZVn7YucgI8f9vrjA/view?usp=sharing). We also highlight the corresponding changes in the submitted revised paper.
>
> 5: The poor perceptual quality of DLL2 is not in agreement with the highest SSIM score in Table 1. DLL2 exhibits and would mislead the reader in terms of the reconstruction of details.
>
> R1.5: We thank the reviewer for pointing this out. The poor perceptual quality of DLL2 is largely due to the “shadow” surrounding the cranial hemorrhage, something not encoded very well in the SSIM measure. In order to quantify the shadow artifact surrounding the hemorrhage, we used the artifact reduction quantification metric used in [1] and modify it to utilize the ground truth (GT): $R_{ICH} = (SD(\chi_{recon}|M_{non-ICH}) - SD(\chi_{GT}|M_{non-ICH})) / SD(\chi_{GT}|M_{non-ICH})$, where $\chi|M_{non-ICH}$ denotes the susceptibilities in the non-ICH region defined as 5-mm-wide layer surrounding each ICH. Therefore, smaller $R_{ICH}$ indicates less shadow artifact surrounding ICH. Results are shown in our updated Table 1 (https://drive.google.com/file/d/13sD30Ey-IN90ZTTOZVn7YucgI8f9vrjA/view?usp=sharing) and Table 2 (https://drive.google.com/file/d/1LzPnS10iXWyUP0_tgRrJ-kmRloUvxbUD/view?usp=sharing). As shown in new Table 1, DLL2 and HOBIT ($\alpha = 1.0, \rho = 60$) have large $R_{ICH}$, which is consistent with heavy shadow artifacts (red arrows) shown in Figure 2a. We also highlight the corresponding changes in the submitted revised paper.
>
> 6: It is not clear if HFEN is represented as %. If so, it must be mentioned.
>
> R1.6: We thank the reviewer for pointing this out. The reviewer is correct that HFEN is defined as RMSE (relative mean square error) between high pass filtered reference and reconstructed susceptibility maps. We update Table 1 and 2 to include the unit of both RMSE and HFEN (%) in the revised paper.
>
> 7: FINE, PDI-VI and HOBIT produced qualitatively better QSMs than the other methods. ----- In that case, the PSNR/SSIM metrics for these images should be provided to understand which method performs better.
>
> R1.7: We add PSNR and $R_{ICH}$ metrics for these images. Please see our updated Figure 3 (https://drive.google.com/file/d/19G-_xVKvjZHDYRjE03FLf49aE5YL8KZz/view?usp=sharing).
>
> 8: In page 2, in the paragraph that starts with “Compared to conventional methods,”, the last two sentences in this paragraph talk about FINE and PDI. To ensure continuity, it would be better to mention about PDI and then lastly talk about FINE, since the following paragraph is about FINE.
>
> R1.8: We thank the reviewer for this suggestion. We modified the manuscript accordingly.
>
> 9: In equation 3, the summation should be i=1 to Nc instead of i to Nc. Similar change must be made in equation 4.
>
> R1.9: These changes have been made.
>
> [1] Liu, Zhe, et al. "Preconditioned total field inversion (TFI) method for quantitative susceptibility mapping." Magnetic resonance in medicine 78.1 (2017): 303-315.

---

### Meta-Review · Area_Chairs · 2021-03-31

**Recommendation:** Accept (Oral)

**Metareview:**

strong paper with many positive remarks, especially after rebuttal. In addition the authors released their code. Recommendation for accept as oral.

**Paper Type:**

methodological development

---

### Decision · Program_Chairs · 2021-03-31

**Decision:**

Accept

**Comment:**

Congratulations your paper has been selected as a long oral.